# Exploring the Quality and Application Potential of the Remaining Tea Stems after the Postharvest Tea Leaves: The Example of Lu’an Guapian Tea (*Camellia sinensis* L.)

**DOI:** 10.3390/foods11152357

**Published:** 2022-08-06

**Authors:** Wenjun Zhang, Wenli Guo, Changxu He, Meng Tao, Zhengquan Liu

**Affiliations:** 1State Key Laboratory of Tea Plant Biology and Utilization, Anhui Agricultural University, Hefei 230036, China; 2School of Tea & Food Science and Technology, Anhui Agricultural University, Hefei 230036, China

**Keywords:** Lu’an Guapian tea, tea stem, tea leaves, growth period, physicochemical properties

## Abstract

Lu’an Guapian tea is produced through the processing of only leaves, with the stems and buds discarded, but stems constitute a large proportion of the tea harvest. To test the usability of tea stems, we compared the physicochemical properties of tea leaves and stems from the same growth period as well as the taste of their infusions. The leaves contained higher concentrations of polyphenols and caffeine and had a stronger taste. The tea stems contained higher concentrations of free amino acids and soluble sugars and were richer in umami and sweet flavors. In addition, more tender tea stems had higher concentrations of polyphenols, caffeine, and free amino acids, and their infusions had more refreshing and sweeter tastes. Furthermore, crude fiber content increased as stem tenderness decreased. In summary, tea stems are rich in phytochemical components and flavor, and these properties increased with tenderness. This provides a theoretical basis for the high-value utilization of tea stems.

## 1. Introduction

Tea (*Camellia sinensis*), a traditional beverage that originated in China, has a wide range of consumers among different ages, places, cultures, societies, etc., due to its unique flavor and healthy benefits [1]. Green tea, the most widely consumed tea in China, can benefit human health because of its chemical composition, which includes atechins, theanine, and caffeine [2,3,4,5].

Tea leaves are plucked along with stems, which constitute a large proportion of plucked tea (approximately 35%). Most teas in China are produced by plucking and processing tea leaves and stems together. Lu’an Guapian (LAGP) tea is one of the 10 most popular teas in China [1]; it is a special single-origin green tea made from only leaves [6]. In the middle of April, when a top bud and four to five leaves grow on a tea stem, the leaves are plucked, leaving the top bud and two leaves so that the stem continues growing. New tea leaves are continually plucked from tea stems, which grow 20–40 cm by the end of the harvesting season. After the tea harvesting season, the stems are cut to avoid nutrient depletion [7]. These stem cuttings are mostly considered unusable biomass and discarded by farmers.

Roasted tea stems are suitable raw materials for ready-to-drink beverages because of their umami and sweet tastes and flavor stability after ultrahigh-temperature (UHT) sterilization. However, the maturity of stems varies from top to bottom. As reported by Xu, maturity is a key factor that determines the quality of tea leaves [8]. Thus, appropriate maturity may also benefit the stems selected for processing. 

The chemical compositions of tea leaves determine the properties of tea, including the color of its infusion and concentrations of polyphenols, caffeine, free amino acids, and soluble sugars. Tea stems also possess numerous physicochemical properties. Lee [9] discovered that the concentrations of theanine, quinic acid, and sucrose in tea stems are significantly higher than those in tea leaves, whereas the concentrations of (−)-epigallocatechin gallate (EGCG) and caffeine are considerably higher in leaves than in stems. In addition, less tender leaves have lower theanine and caffeine content and higher sucrose content. Yamashita [10] revealed that the theanine content of new stems is significantly higher than that of new leaves, and the catechin content of new leaves is significantly higher than that of new stems. Although leaves typically have higher concentrations of phytochemical components and better quality compared with stems, a few researchers have focused on tea stems, including those of LAGP and oolong tea and those separated during tencha processing. To test the applicability of tea stems, we compared the taste and phytochemical composition between LAGP leaves and stems with the same maturity and among LAGP stems with different maturity.

## 2. Materials and Methods

### 2.1. Samples and Chemicals

Tea samples (*Camellia sinensis* cv. local varieties) were collected in mid-May 2021 from Mabu Town, Jinzhai County, Lu’an City, Anhui Province, China (31° N, 116° E). As illustrated in Figure 1, the tea stems were severed from plucked tea leaves, the reddish-brown stem (fresh tea stem 6 [F-TS6]) was set aside, and the remaining stems were divided equally into fresh tea leaf and stems 1 through 5 (F-TLS1–F-TLS5). 

The F-TLS1 sample was processed to obtain tea leaf 1 (TL1) and tea stem 1 (TS1). The final samples obtained after processing were TL1, TS1, tea stem 2 (TS2), tea stem 3 (TS3), tea stem 4 (TS4), tea stem 5 (TS5), and tea stem 6 (TS6). All of samples were processed with the same parameters (fixation at 200 °C for 16 min, drying at 110 °C for 50 min) and stored at −20 °C.

High-performance liquid chromatography (HPLC)-grade acetonitrile, methanol, and acetic acid were procured from Tedia (Fairfield, OH, USA). Samples of caffeine, EGCG, (−)-gallocatechin gallate (GCG), (−)-epicatechin gallate (ECG), (−)-epigallocatechin (EGC), (−)-gallocatechin (GC), (−)-epicatechin (EC), and (+)-catechin (C) were procured from Sigma (St. Louis, MO, USA). Sixteen amino acids (aspartic acid, serine, glutamic acid (Glu), glycine, histidine, arginine, threonine, alanine, proline, cysteine, tyrosine, valine, lysine, isoleucine, leucine, and phenylalanine) were purchased from Waters (Milford, MA, USA). L-theanine (Thea) was purchased from Solarbio (Beijing, China). An AccQ-Tag-Fluor Reagent Kit and AccQ-Tag-Eluent A Concentrate were also purchased from Waters (Milford, MA, USA). All the chemicals and solvents used for the extraction of samples were of analytical grade.

### 2.2. Tea Leaf and Stem Quality Analysis

#### 2.2.1. Color Differences between Tea Leaves and Stems and Their Infusions 

The colors of the dry samples were analyzed using a color difference meter (Color Quest XE) in reflected light mode in accordance with Huang’s method [11]. L* indicates brightness, a* indicates redness/greenness, and b* indicates yellowness/blueness; specifically, a larger L* value indicates that a sample is brighter, −a* indicates the greenness of the sample, +a* indicates the redness of the sample, −b* indicates blueness, and +b* indicates yellowness. The colors of the infusions were determined according to the parameters of the light passing through them, using the color difference meter in transmitted light mode. The infusions were prepared through the national standard brewing method [12], with 3 g of each tea sample brewed in 150 mL boiling water for 5 min and then removed from the broth before the infusion was filtered.

#### 2.2.2. Tea Taste and Quality 

The samples were analyzed for sensory quality on the basis of a version of Chinese standard GB/T23776-2018 [12] modified to suit the characteristics of the samples. After training, six (three male and three female) panelists scored the samples (0–10) for bitterness, astringency, sweetness, umami, and strength. This analysis was performed in triplicate.

### 2.3. Detection of Major Chemical Compounds

#### 2.3.1. Catechin, Caffeine, and Total Polyphenol Content

The method used to detect caffeine and catechins in this study was based on previously reported methods [13,14,15]. The samples were extracted through the method described by Ning [16], with minor modifications; 0.2 g of tea powder (obtained using a grinder; it can pass through a 20 mesh screen) was extracted twice using 5 mL of 70% methanol to obtain 10 mL of a solution for assaying. The obtained solution was filtered through a 0.22 µm nylon membrane and analyzed through HPLC. Chromatographic analysis was performed using a previously reported method [17], with slight variations. HPLC was performed using a high-performance liquid chromatograph (Agilent Technologies, Palo Alto, CA, USA) equipped with a chromatography column (250 × 4.6 mm^2^, 5 μm, Phenomenex Synergi Hydro-RP C18 column). Mobile phase A was composed of 9% acetonitrile, 2% acetic acid, and 0.2% ethylenediaminetetraacetic acid (EDTA), and mobile phase B was composed of 80% acetonitrile, 2% acetic acid, and 0.2% EDTA. A linear gradient elution program was as follows: 0–10 min, 100% A; 10–25 min, 100% to 68% A and 0% to 32% B; 25–35 min, 68% A and 32% B; 35–40 min, 68% to 100% A and 32% to 0% B; and 40–60 min, 100% A. The flow rate was 1 mL/min, and the detection wavelength was 278 nm. The catechins and caffeine were detected through the comparison of the retention times and ultraviolet spectra with standard solutions. Calibration plots were created for caffeine, EGC, C, EC, EGCG, and ECG (Appendix A). The concentration of gallated catechins was calculated using the following equation [18]: Gallated catechins (%) = EGCG% + ECG%.

The concentration of total tea polyphenols was determined according to Chinese standard GB/T 8313-2018 [19]. First, 1 mL of a solution to be tested for catechins was diluted and diluted 100 times; then, 5 mL of 1 mol/L Folin–Ciocalteu reagent was added to 1 mL of the diluted solution. After the solution and reagent reacted for 3–8 min, 4 mL of 75% Na_2_CO_3_ was added; water was added to a fixed volume, and then the mixture was shaken well and left for 60 min at room temperature. The absorbance at 765 nm was measured using a spectrophotometer. A standard gallic acid curve was plotted to evaluate the total polyphenol content (Appendix A).

#### 2.3.2. Amino Acid and Free Amino Acid Content 

Various amino acids were detected through a method presented by Boogers [20] on an Agilent 1260 HPLC system (Agilent Technologies, Palo Alto, CA, USA) with appropriate modifications. The oven temperature was set at 37 °C. Various amino acids (and L-theanine) were detected at 248 nm at a flow rate of 1.0 mL/min. Mobile phase A comprised 9.09% (1/11) acetonitrile and 90.01% ultrapure water, and mobile phase B comprised 60% acetonitrile and 20% ultrapure water. The following gradient elution program was employed: 0 min, 100% A; 0–0.5 min, 100% to 98% A and 0% to 2% B; 0.5–15 min, 98% to 93% A and 2% to 7% B; 15–19 min, 93%–90% A and 7% to 10% B; 19–35 min, 90% to 67% A and 7% to 33% B, maintained for 1 min; 36–37 min, 67% to 0% A and 33% to 100% B, maintained for 5 min; and 42–43 min, 0% to 100% A and 100% to 0% B, maintained for 7 min. The scalar curves for the individual free amino acids are displayed in Appendix A.

L-theanine content was determined according to Chinese standard GB/T23193-2017 [21]. The oven temperature was set at 35 °C. L-theanine was detected at 210 nm and a flow rate of 1.0 mL/min. Mobile phase A was composed of 100% ultrapure water, and mobile phase B was composed of 100% acetonitrile. The following gradient elution was performed: 0–10 min, 100% A and 0% B; 10–12 min, 100% to 20% A and 0% to 80% B, maintained for 8 min; and 20–22 min, 20% to 100% A and 80% to 0% B, maintained for 18 min. Calibration plots were obtained for L-theanine (Appendix A).

Free amino acid content was determined according to Chinese standard GB/T 8314-2013 [22]. First, 3 g of a crushed sample was added to 450 mL and extracted for 45 min. The extract was filtered, cooled, and fixed at 500 mL. Then, 1 mL of the extract was added to 0.5 mL of phosphate buffer solution with a pH of 8.0 and 0.5 mL of 2% ninhydrin solution in a boiling water bath, where it was maintained for 15 min; the mixture was cooled and fixed at 25 mL with water and allowed to rest for 10 min. The absorbance at 570 nm was measured. A standard curve for theanine was plotted to evaluate the free amino acid content (Appendix A). The phenol–ammonia ratio was calculated using the following equation [23]: Phenol–ammonia ratio = (Total polyphenols (%, dw))/(Total free amino acids (%, dw)).

#### 2.3.3. Monosaccharide, Disaccharide, and Total Soluble Sugar Content 

Sugar content was detected using an Agilent 8890-5977B platform. Sample pretreatment was performed through the methods reported by Gomez-Gonzalez, Medeiros, and Zheng et al [24,25,26]. The Agilent 8890 gas chromatograph coupled to a 5977B mass spectrometer with a DB-5MS column (30 m (length) × 0.25 mm (inner diameter) × 0.25 μm (film thickness), J&W Scientific, USA) was used to perform gas chromatography–mass spectrometry of sugars. Helium at a flow rate of 1 mL/min was used as the carrier gas. Injections were made in split mode with a split ratio of 5:1 and an injection volume of 1 μL. The oven temperature was maintained at 170 °C for 2 min; raised to 240 °C at 10 °C/min, to 280 °C at 5 °C/min, and to 310 °C at 25 °C/min; and finally the temperature was maintained for 4 min. All samples were analyzed in selective ion-monitoring mode. The ion source and transfer line temperature were 230 °C and 240 °C, respectively [25,27]. Calibration plots were constructed for monosaccharide and disaccharide content (Appendix A).

The total soluble sugar content of each sample was determined according to the method reported by Ning [28] with a few modifications. First, 80 mL of boiling water was added to 1 g of the sample, which was maintained in a boiling water bath for 30 min; the extract was filtered, cooled, and fixed at 500 mL. Then, 1 mL of the extract was added dropwise to a reagent containing 8 mL of anthrone; distilled water was used as a control. The mixture was shaken well and placed in a boiling water bath for 7 min; then it was immediately transferred to an ice water bath at room temperature. The absorbance at 620 nm was measured. A standard glucose curve was plotted to evaluate the total soluble sugar content (Appendix A).

#### 2.3.4. Water-Soluble Substance, Crude Fiber, and Ash Content 

The water-soluble substance content of the samples was determined according to Chinese standard GB/T 8305-2013 [29]. First, 2 g of a crushed sample was added to 300 mL of boiling water and maintained in the boiling water bath for 45 min (shaken once for 10 min), filtered while still hot, cooled, and filtered again. The water-soluble substance was evaluated as the difference in mass between the dry tea residue and dry matter content of the leaves before infusion.

The crude fiber content of the samples was determined according to Chinese standard GB/T 83-2013 [30], with appropriate modifications. Each sample was subjected to acid and alkaline digestion with a cleaned crude fiber tester. Then, 2.5 g of the sample was crushed and subjected to acid digestion using a 1.25% H_2_SO_4_ solution for 30 min and to alkaline digestion using a 1.25% NaOH solution for 30 min. The sample was then decolorized using ethanol and acetone, dried at 120 °C for 2 h, placed in a muffle furnace at 300 °C until the smoke was exhausted, and finally heated at 500 °C for 30 min until ash was formed. 

Ash content was determined according to Chinese GB5009.4-2016 [31]. First, 3 g of each crushed sample was weighed and heated before being charred in an electric oven until the smoke was exhausted; the charred sample was placed in a muffle furnace and baked at 550 °C ± 25 °C for 4 h, cooled to 200 °C, transferred to a drying dish, and cooled for 30 min. The ash remaining in the crucible was weighed. The ash was extracted using hot water, filtered through ash-free filter paper and scorched. The residue, that is, the water-insoluble ash, was weighed. The mass of the water-soluble ash was considered the difference between the mass of the total ash and water-insoluble ash. These experiments were performed in triplicate.

### 2.4. Statistical Analysis 

Analysis data are expressed as means ± standard (replicate samples of each type and repeated three times) variances. A *p* value of <0.05 was considered to indicate a significant difference. Significant differences between samples were analyzed using SPSS (version 25.0; SPSS, Chicago, IL, USA). Heat maps were produced using TBtools (version JRE1.6). Principal component analysis (PCA), hierarchical cluster analysis (HCA), and load mapping were performed using SIMCA (version 14.1; MKS Umetrics, Umea, Sweden).

## 3. Results

### 3.1. Differences in Quality between Leaves and Stems

#### 3.1.1. Color Differences between Tea Leaves and Stems and Their Infusions 

Figure 2a displays the dried tea stem samples, which were considerably brighter and yellower than the leaves with the same degree of tenderness. However, the greenness of the stems was significantly lower than that of the leaves, most likely because leaves contain more chlorophyll [32], which imparts a darker and greener color. As illustrated in Figure 2b, the tea infusion prepared using the stems was slightly brighter, optically rarer, and yellower, whereas the tea infusion prepared using the leaves was slightly darker and greener. According to previous reports, this can be attributed to the higher quercetin and chlorophyll content of leaves [33]. 

As illustrated in Figure 2c, the leaves were more bitter and astringent than the stems with the same tenderness, whereas the stems exhibited more umami and sweet flavor. Furthermore, the infusions prepared using the leaves were stronger than those prepared using the stems, likely because of the higher abundance of metabolites—especially of gallated catechins and caffeine, which contribute to the bitter taste [34]—and because of the higher free amino acid and sugar content of the stems. 

#### 3.1.2. Differences in Color and Taste among Tea Stems with Different Degrees of Tenderness

As illustrated in Figure 3a, less tender dried stems had less greenness but more yellowness and brightness. Notably, the stem bark of woody plants also contains chlorophyll [35]. The chlorophyll content of tea stems may decrease with tenderness, thereby increasing the yellowness and brightness of the tea stems. As depicted in Figure 3b, the older the stems were, the brighter, lighter, and less green their infusion was, likely because older tea stems have lower quercetin and chlorophyll content. Figure 3c illustrates that the older stems exhibited a weaker taste and less freshness and sweetness, likely because they had lower metabolite content, especially that of free amino acids. This may also be related to the low leaching rate of older tea stems [36,37]. However, TS2 had a stronger flavor than TS1 had, likely because TS2 had higher total catechin and EGCG content [38].

### 3.2. Differences in Primary Physicochemical Properties 

#### 3.2.1. Differences in Primary Physicochemical Properties between Tea Leaves and Stems

The primary chemical compositions of the tea leaves and stems with the same degree of tenderness are illustrated in Figure 4. The experimental results indicated that the water extract, total polyphenol, caffeine, EGC, EGCG, ECG, nongallated catechin, total catechin, and water-insoluble ash content of the leaves was considerably higher than that in the stems (*p* ≤ 0.05). However, the total free amino acid, crude fiber, C, EC, water-soluble ash, and ash content of the stems was higher than that of the leaves (*p* ≤ 0.05). The phenol–ammonia ratios of the leaves were considerably higher than those of the stems. The levels of soluble sugar and total ash in the stems were slightly higher than those in the leaves. These results could be attributed to the higher tryptophan and indoleacrylic acid content of leaves, which contribute to more active secondary metabolism and leaf growth. However, stem accumulation of quinic and shikimic acids contributed to stem lignification. Furthermore, the stems accumulated L-theanine, pyroglutamic acid, 4-aminobutyric acid, and glutamine, which may promote stem elongation [39].

#### 3.2.2. Differences in Primary Phytochemical Properties among Tea Stems with Different Degrees of Tenderness

As tenderness decreased, the water-soluble substance, total polyphenol, free amino acid, C, EC, nongallated catechin, water-soluble ash, water-insoluble ash, and total ash content gradually decreased (Figure 5). Aside from TS2 having higher caffeine, EGC, EGCG, ECG, gallated catechin, and total catechin content than TS1, the caffeine and aforementioned catechin content decreased thereafter with tenderness. The phenol–ammonia ratio and soluble total sugar content exhibited fluctuating but downward trends as tenderness decreased, whereas the crude fiber content of the tea stems increased. These trends could be attributed to the conversion of the various internal constituents of the stems with decreasing tenderness to promote the production of substances such as cellulose [40] and enhance mechanical support [41].

#### 3.2.3. Differences in the Various Amino Acid, Monosaccharide, and Disaccharide Content of Tea Leaves and Stems 

As illustrated in Figure 6a, the concentrations of the free amino acid components of the leaves were generally lower than those of the stems, and the free amino acid components of the stems exhibited an overall decreasing trend with decreasing tenderness. Among the free amino acids in the samples, theanine was the most abundant, accounting for 25–50% of the total free amino acids. The monosaccharide and disaccharide content of the leaves was generally lower than that in the stems, which exhibited a fluctuating but downward trend with decreasing tenderness. Among the monosaccharides and disaccharides in the samples, sucrose was the most abundant, accounting for 30–50% of the total soluble sugars. The stems had significantly more theanine and sucrose than the leaves (Figure 6b). The more tender the tea stems were, the greater the theanine content they had. Theanine content may decrease with the tenderness of tea stems because of the enzymatic breakdown of theanine into Glu and ethylamine increases as stems mature [42]. This process increases not only sweetness but also amino acid content and facilitates the generation of heterocyclic aromatic compounds in roasting processes [43,44], thereby improving the flavor of the tea and its robustness to UHT sterilization. 

### 3.3. Multivariate Analysis 

To rapidly and effectively compare tea stems of different maturity, PCA was first used to provide an overview of all the aforementioned chemical data. PCA is an unsupervised multivariate statistical analysis method based on data dimensionality reduction [45], which can objectively reflect the information provided by the original variables for linear classification. As illustrated in Figure 7A, principal components 1 and 2 accounted for 83.35% and 11.93%, respectively, of the information provided by the samples (95.28% total), thus indicating that the PCA model comprehensively reflected the data. PCA was performed on the basis of 47 original variables: water-soluble substance, total free amino acid, total tea polyphenol, caffeine, EGC, C, EGCG, EC, ECG, total catechin, gallated catechin, nongallated catechin, crude fiber, total soluble sugar, water-soluble ash, water-insoluble ash, total ash, 16 amino acids, 4 disaccharides, and 9 monosaccharides. Figure 7A,B display the PCA scores and loading plots, respectively. The seven samples were divided into three groups: The first group contained TL1; the second group contained TS1, TS2, and TS3; and the third group contained TS4, TS4, and TS6. The main differences between the first group and the second and third groups were gallated catechins and some monosaccharides. The key differences between the second and third groups were those in crude fiber, some amino acids, and especially theanine. As depicted in Figure 7A, obvious differences were noted between the leaves and stems, reflecting the differentiation and gradual changes tea stems undergo during their growth. For HCA, the seven samples were divided into three groups, and the results were consistent with those of the PCA, as illustrated in Figure 7C. On the basis of the PCA and HCA results, the tea stems were roughly divided into two groups. The stems with higher tenderness (Group 3) and more beneficial components, can be used to extract substances such as theanine or processed to produce tea beverages. The feasibility of such processes was determined through further experimentation. The processing of tea stems according to their various components may be a direction for achieving high value from tea stems.

## 4. Conclusions

This study discovered significant differences in the composition of tea leaves and stems, with higher levels of catechins and caffeine in leaves and higher levels of free amino acids and soluble sugars in stems. These differences result in distinct flavor profiles in infusion, with the leaf infusion being more bitter and astringent and the stem infusions being sweeter and richer in umami. As the tenderness of tea stems decreases, the levels of most flavor compounds decrease, and cellulose content considerably increases; furthermore, the sweetness and strength of the infusion gradually decrease. The results of this study indicate the differences between tea leaves and stems and demonstrate the value of tea stems, especially tender ones. Furthermore, the results regarding changes in the composition of tea stems with tenderness provide a theoretical basis for the efficient use of tea stems, especially those typically discarded after LAGP harvesting.

## 5. Patents

This section is not mandatory but may be added if there are patents resulting from the work reported in this manuscript.

## Figures and Tables

**Figure 1 foods-11-02357-f001:**
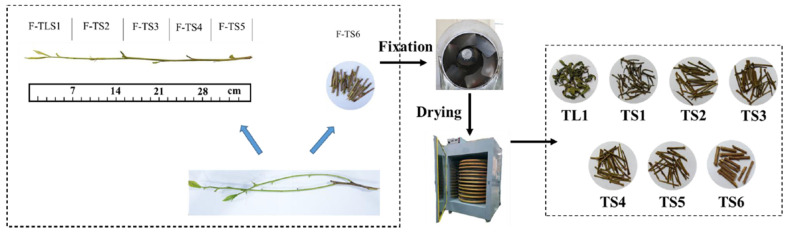
Processing of tea leaf and stem samples.

**Figure 2 foods-11-02357-f002:**
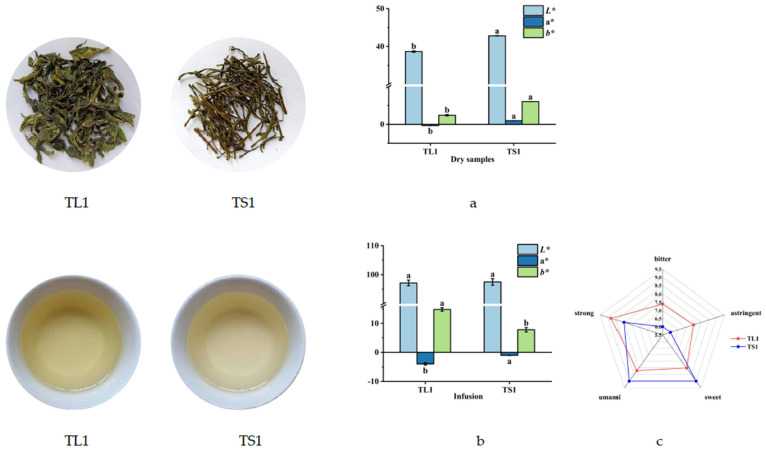
Difference in the colors of dry leaf and stem samples and infusions and sensory evaluation of stems and tea leaf infusions. (**a**) Difference in the colors of dry leaf and stem; (**b**) Difference in the colors of stems and tea leaf infusions; (**c**) Sensory evaluation of stems and tea leaf infusions. Different superscript letters denote significant differences (*p* < 0.05).

**Figure 3 foods-11-02357-f003:**
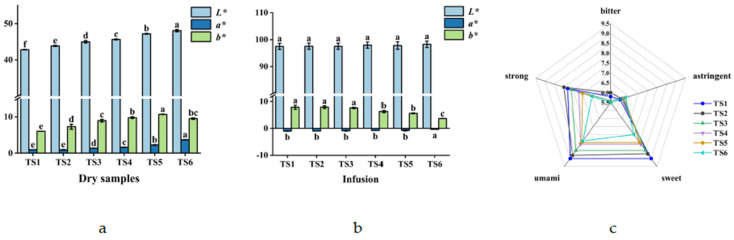
Differences in color between dry stem samples and stem infusions and sensory evaluation of stem infusions. (**a**) Difference in the colors of dry stems; (**b**) Difference in the colors of stems infusions; (**c**) Sensory evaluation of stems infusions. Different superscript letters denote significant differences (*p* < 0.05).

**Figure 4 foods-11-02357-f004:**
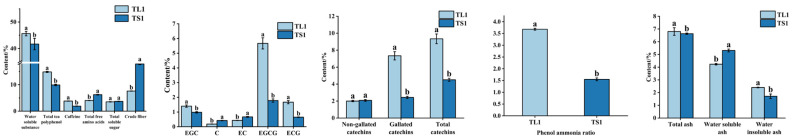
Primary phytochemical composition of tea stems and leaves with the same degree of tenderness. Different superscript letters denote significant differences (*p* < 0.05).

**Figure 5 foods-11-02357-f005:**
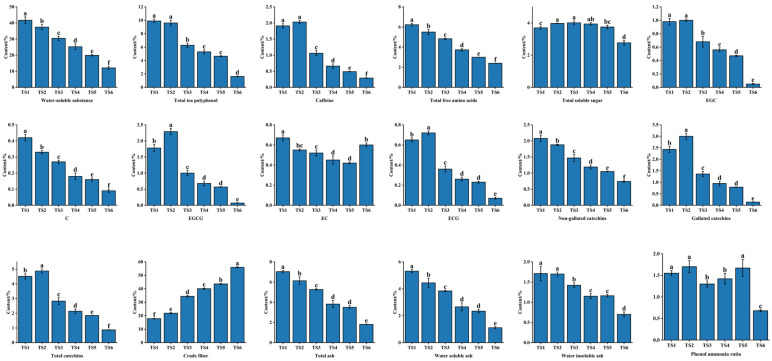
Primary phytochemical composition of tea stems with different degrees of tenderness. Different superscript letters denote significant differences (*p* < 0.05).

**Figure 6 foods-11-02357-f006:**
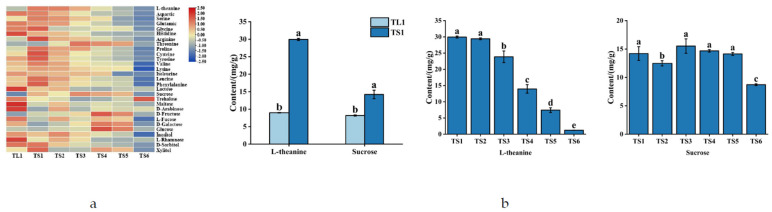
Free amino acids, monosaccharides, and disaccharides in tea leaves and stems. (**a**) Heatmap of free amino acids, monosaccharides, and disaccharides in tea leaves and stems; (**b**) L-theanine and sucrose in tea leaves and stems. Different superscript letters denote significant differences (*p* < 0.05).

**Figure 7 foods-11-02357-f007:**
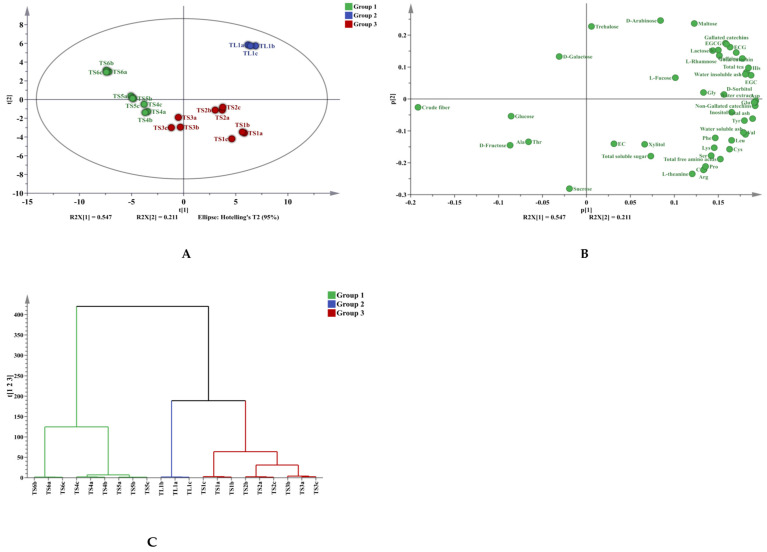
PCA and HCA analysis of tea leaves and tea stems. (**A**) PCA analysis of tea leaves and tea stems; (**B**) Loading plots; (**C**) HCA analysis of tea leaves and tea stems.

## Data Availability

Data is contained within the article.

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
