# Peer review of "Exploring the Quality and Application Potential of the Remaining Tea Stems after the Postharvest Tea Leaves: The Example of Lu’an Guapian Tea (*Camellia sinensis* L.)"

_foods, 2022, doi:10.3390/foods11152357_

Round 1
Reviewer 1 Report
Exploring the quality and application potential of the remaining tea stems after the ’postharvest tea leaves: the example of Lu'an Guapian tea (Camellia sinensis L.)
The paper reports a study on the difference in some properties of leaves and steams of Lu'an Guapian tea to discover a possible utilization of tea stems usually discarded as unusable biomass.
The work features enough novelty, and it could be of interest for recycling industry for a possible recovery of commonly unused waste material.
The manuscript requires some language editing; please check the English meaning of all the manuscript also with the help of a mother language speaking.
SUGGESTED CORRECTIONS
ABSTRACT
Line 19: I can't understand what you mean with “physicochemical components”. The components of the matter are chemical. Maybe you can substitute “physicochemical” with “nutraceutical” or “phytochemical”.
Line 20: Eliminate “are”
1. INTRODUCTION
Line 30: Eliminate the dot after “plucked”
Line 40: As before. The composition is chemical and cannot be physical. Eliminate “physical and”
Line 50: “physicochemical components” as before.
Line 53: As before. Eliminate “physicochemical”.
2. MATERIALS AND METHODS
Line 53: Specify the year of harvest
Lines 86-87: “according to the parameters 86 of the light passing through them” Please explain better
Line 88: “standard brewing method” Please add reference
Line 88: “boiling water” Please specify the quantity
Lines 91-92: “Chinese standard GB/T23776-2018” Please add reference
Lines 116-117: “Chinese standard GB/T 8313-2018” Please add reference
Lines 117-118: Eliminate “aspirated and”
Line 120: “Na2CO3”: subscript the numbers Na2CO3
Line 136: “Chinese standard GB/T23193-2017” Please add reference
Lines 143-144: “Chinese standard GB/T 8314-2013” Please add reference
Line 174: Define the "Water extract" as "Water-soluble solid" or "Water-soluble substance"
Line 175: See line 174
Lines 175-176: “Chinese standard GB/T 8305-2013” Please add reference
Line 178: See line 174
Line 179: Add “dry” before “tea residue”
Lines 180-181: “Chinese standard GB/T 83-2013” Please add reference
Line 183: “H2SO4”: subscript the numbers H2SO4
Line 187: “Chinese GB5009.4-2016” Please add reference
3. RESULTS
Lines 212-213: “According to previous reports, this can be attributed to the higher flavonoid content of leaves [20].”: Reference "20" report that the greenness can be correlated to quercetin and chlorophyll content. "Flavonoid" is a too general term. Please change it
Line 229: "Flavonoid": see lines 212-213
Figure 2 Caption: Add “stem” after “dry”
Line 236: “physicochemical” only chemical properties are described
Line 237: “physicochemical” see line 236
Lines 239-249: "Water extract" see line 174
Lines 246-251: “These results could be attributed to the higher chlorophyll content of leaves [19] which 246 provides energy for synthesizing more chemical compounds. However, stems behave as 247 conduits for nitrogen, transporting the nitrogen absorbed by the roots to the leaves [26], 248 and have more abundant nitrogen and higher free amino acid content. Furthermore, stems 249 support leaves and require high mechanical strength, which is endowed by their high 250 crude fiber.”
I am not very convinced of these explanations and the references given do not support them.
Figure 4: Change "Water extract" as stated in line 174 in the axis description of the first graph
Figure 4 Caption: “physicochemical” see line 236
Line 254: “physicochemical” see line 236
Line 256: "Water extract" see line 174
Line 258: Substitute "Other than" with "Aside from"
Figure 5: Change "Water extract" as stated in line 174 in the axis description of the first graph
Figure 5 Caption: “physicochemical” see line 236
Line 289: “physical and” All the data used for PCA analysis are chemical
Line 295: "Water extract" see line 174
Line 302: "esterified” or “gallated” instead of “ester”
5. PATENTS: Remove this section

Author Response
非常感谢您的宝贵意见。有关我的回复的详细信息,请参阅附件。

Reviewer 2 Report
The manuscript is mostly well-written and the Results and Discussion section is clear and logically organised. A few things need to be clarified in the Materials and Methods section (see annotated pdf).

Author Response
Thank you very much for your valuable comments. Please see the attachment for details of my response.

Reviewer 3 Report
Dear Authors,
The manuscript requires significant revision in terms of introduction, results and discussion.
Title : Exploring the quality and application potential of the remaining tea stems after the ’postharvest tea leaves: the example of 3 Lu'an Guapian tea (Camellia sinensis L.)
Authors: Wenjun Zhang, Wenli Guo, Changxu He, Meng Tao and Liu Zhengquan
- Title: The title of the manuscript accurately reflects the content of the paper.
- Abstract and Key words:
The abstract lacks information about the tea classification method. Key words are adequate to the content of the manuscript.
- Introduction: The introduction requires revision. The reference to 4 items of literature is definitely not enough. Please extend the introduction.
- Objectives and hypotheses: The aim of the study was clearly stated. The hypotheses was accurately connected with state-of-knowledge in the discussion section.
- Methods: adequate to the aims of the study.
- Results and Discussion:
The results are presented in 3 tables and attached to the appendix. I believe that the tables should be included in the manuscript and not in a separate appendix. There are references in the manuscript to figures that are not there. Apart from that, it is a poor material when it comes to presenting the results. Complete no discussion in the manuscript. Figures and tables: requires correction-completion.
- Abbreviations, formulae, units: conform to acceptable standards.
- Literature cited: relevant.
Best Regards
Author Response

(The authors gave the same response as above.)

Round 2
Reviewer 3 Report
Dear Authors,
thank you for editing the work and introducing significant changes.
I recommend the manuscript for publication.
Best Regards